# Thromboembolic Events in a Socio-Economically Disadvantaged Population with COVID-19 Admitted to a Medicalized Hotel in Madrid

**DOI:** 10.3390/ijerph19137816

**Published:** 2022-06-25

**Authors:** Karen Lizzette Ramírez-Cervantes, Consuelo Huerta-Álvarez, Manuel Quintana-Díaz

**Affiliations:** 1Department of Disease Prevention and Health Promotion, Spanish Association Against Cancer, 28040 Madrid, Spain; 2Patient Blood Management Research Group, Research Institute of La Paz University Hospital, 28040 Madrid, Spain; mquintanadiaz@gmail.com; 3Department of Public Health & Maternal and Child Health, Faculty of Medicine, Complutense University of Madrid, 28040 Madrid, Spain; mahuer05@ucm.es; 4Intensive Care Unit, La Paz University Hospital, 28046 Madrid, Spain

**Keywords:** thrombosis, thrombotic events, coronavirus disease 2019, COVID-19, socio-economically disadvantaged population, social determinants of health

## Abstract

**Background:** The social determinants of health (SDOH) of patients with COVID-19-related thrombosis have been scarcely explored. Our objective was to investigate the cases of thrombosis in a group of socially disadvantaged populations with COVID-19. **Methods:** We investigated the thrombotic events that occurred in a cohort of migrant and Spanish patients with COVID-19 that were admitted to a medicalized hotel in Madrid. Demographic data, past medical history, and socio-economic backgrounds, such as monthly household income, level of education, and living conditions, were explored to determine the factors related to thrombosis. **Results:** A cohort of 383 subjects (mean age 55.4 ± 14.6 years old, 69% male), of which 58% were migrants, was studied. Fourteen (3.6%) cases of thrombosis were reported. Thrombosis was more frequent in Spanish than in migrant individuals (OR 5.3, 95%CI 1.4–19.5, *p* = 0.005). Neither a low monthly household income nor a low education level showed a statistical association with thrombosis (*p* ≥ 0.05). History of venous thromboembolism (OR 8.1, 95%CI 2.2–28.6) and being a current smoker (OR 4.7, 95%CI 1.3–16.0) were factors associated with thrombosis. **Conclusions:** The SDOH studied were not associated with thrombosis; however, further investigation must be performed to investigate the socio-economic conditions of subjects with COVID-19 with adverse outcomes such as thrombotic events.

## 1. Introduction

Abnormal coagulation parameters and thrombotic events, such as pulmonary embolism (PE), deep vein thrombosis (DVT), stroke, and acute myocardial infarction (AMI), have been widely described among patients with coronavirus disease 2019 (COVID-19) [1,2,3]. Certain host factors, such as stasis of circulation, vascular endothelial injury, and hypercoagulability, could heighten the risks of thrombosis [4]. However, the determinants of thrombotic events, including those related to social determinants of health (SODH), have not been fully elucidated in subjects with COVID-19 [5].

The hyperinflammatory response observed in COVID-19 individuals could be facilitated by pre-existing non-communicable diseases, such as diabetes; hypertension; cardiovascular diseases; and chronic respiratory, kidney, or liver conditions. However, it is increasingly demonstrable that social background, such as poverty, level of education, physical environment (e.g., housing, occupation, and smoke exposure), race, and ethnicity have a considerable effect on COVID-19 outcomes [6,7]. For instance, socially disadvantaged populations have a higher risk for severe acute respiratory syndrome coronavirus 2 (SARS-CoV-2) and higher COVID-19 mortality rates, as observed in Canada, where nearly 44% of people testing positive for SARS-CoV-2 during the first waves of COVID-19 were migrants. Similarly, in the United States, poverty has shown to be a significant risk for COVID-19-related mortality [8,9,10].

Material deprivation may play an essential role in developing other COVID-19-related outcomes, such as thrombosis, as observed by Kort et al., who reported that in neighborhoods with a higher socio-economic status, a lower risk for venous thromboembolism was found [11]. Likewise, in a cohort of more than 53,000 subjects hospitalized for PE, risk-adjusted 30-day and 90-day all-cause readmission rates were significantly higher in socio-economically deprived individuals (30-day HR, 1.14 [95% CI, 1.06–1.22]; 90-day HR, 1.18 [95% CI, 1.12–1.25]), who also presented higher 1 year mortality rates (hazard ratio [HR], 1.16; 95% CI, 1.10–1.22) [12]. It is, nonetheless, unclear whether, besides the clinical characteristics of COVID-19 individuals, deprived pre-existing social conditions are related to thrombosis. Therefore, the objective of our study was to explore the socio-economic characteristics associated with venous and arterial thrombosis in socially disadvantaged subjects with COVID-19.

## 2. Materials and Methods

We studied the cohort of patients with COVID-19 admitted to the Via Castellana Medicalized Hotel (VCMH) between March and May 2020 [13]. The VCMH was implemented in Madrid after the first wave of cases of COVID-19 as a response to decreasing the spreading of severe acute respiratory syndrome coronavirus 2 (SARS-CoV2), providing shelter for quarantining and medical support to vulnerable populations. Most patients were previously hospitalized and experienced moderate COVID-19. However, further details regarding the characteristics of this population are described elsewhere [13]. Unlike the first report, this study includes an in-depth analysis between migrant and Spanish individuals and an exploratory data analysis of the thrombotic events presented after the COVID-19 diagnosis.

We investigated arterial and venous thromboembolism occurrence within 180 days after COVID-19 diagnosis (study period) in patients admitted to the VCMH. The electronic medical records (EMR) of La Paz University Hospital and the electronic health record systems of primary and specialty care in Madrid (EHRs-Horus) were retrospectively revised. The study period ended before the COVID-19 vaccination program started in Spain. Therefore, when performing this study, none of the patients had been vaccinated against COVID-19.

We confirmed the diagnosis of COVID-19 if a positive real-time reverse transcription-polymerase chain reaction (RRT-PCR) test for SARS-CoV2 in nasopharyngeal swabs was found in the EHRs-Horus, or if a medical doctor reported the diagnosis of COVID-19 in the letter of referral to the VCMH. The index date was the day the first positive test for SARS-CoV2 was registered. Venous thrombosis included PE, DVT, portal vein thrombosis, or intracranial venous thrombosis, whereas arterial thromboembolism included AMI or acute ischemic or embolic stroke [5]. We excluded subjects with incomplete information. In addition, as a medical follow-up was mandatory after the VCMH discharge, we did not include subjects in whom no further medical follow-ups were found in the HER-Horus.

We studied subjects’ characteristics such as age, gender, previous comorbidities, and daily medications reported at the time of COVID-19 diagnosis (baseline information) in both Spanish and migrant individuals. The data concerning hospitalization comprised length of hospital and hotel stays and dose and duration of anticoagulant therapy. We also collected the information on the patients’ socio-economic characteristics, such as average monthly household income, educational level, and living conditions, including living in overcrowding; cohabitating with individuals at high risk for severe illness from COVID-19; and being homeless, evicted, or shelter residents [13]. We considered overcrowding when a household crowding index (HCI) was >2.5. HCI was defined as the total number of inhabitants per house (excluding newborn infants), divided by the total number of rooms in the household, except the kitchen and bathrooms.

### Statistical Analysis

We performed a case-control analysis. Categorical variables were presented as absolute and relative frequencies. In contrast, numeric variables were reported as means and standard deviation (SD) or medians and interquartile range (IQR), according to their respective normal or non-normal distribution. The Shapiro−Wilks test was used to determine the normality of quantitative variables. The comparison between nonparametric numeric data and nominal variables of two categories was performed using the Mann−Whitney U-test, while the chi-square (χ^2^) test was used to compare categorical data. Statistical significance was set at a *p* < 0.05.

We determined the incidence of the primary outcome by calculating the number of patients with a diagnosis of arterial or venous thromboembolism, divided by the overall number of subjects included in the study. The incidence of the primary outcome was reported on days 28, 90, and 180 after the index date. A bivariate analysis was performed to calculate the relative odds of the occurrence of a thrombotic event given the patients’ characteristics. Moreover, we carried out a multivariate analysis with the forward conditional method, introducing thrombosis as the dependent variable and adjusting it with the variables that resulted in being associated with thrombosis (*p*-value < 0.05) in the bivariate analysis. The association between exposure and outcome was represented in a crude or adjusted odds ratio (OR) with a 95% confidence interval (95% CI). We used SPSS software version 21 to perform this analysis.

## 3. Results

We excluded 16 subjects from this analysis. Therefore, a total of 383 individuals (69% males) with a mean age of 55.4 ± 14.6 years old were included (Figure 1). Of them, 161 were migrants and 222 had Spanish nationality (Table 1). Having a low monthly household income (<950 €, according to the minimum interprofessional wage in Spain in 2020) (18% vs. 9%) and living in overcrowding (69% vs. 44%) were characteristics more frequently observed in migrants than in Spanish individuals (Table 1).

The 180day prevalence of thrombosis was 3.6% (14 cases), with men accounting for 71%. Thrombosis was more likely in Spanish than in migrant individuals (OR 5.3, 95% CI 1.4–19.5, *p* = 0.005) (Table 2). In general, the cases of venous thromboembolism were more common than arterial thrombosis, as PE was detected in 86% of subjects (*n* = 12/14), and only one case of iliac artery thrombosis and one case of middle cerebral artery stroke were identified. We observed that most thrombotic events occurred within the first 90 days after the index date (86%, *n* = 12/14), most of them being diagnosed during hospitalization (57%) (Figure 2).

The lengths of stay in the hospital (12.5 ± 7.1 vs. 8 [IQR 5–10], *p* = 0.357) and in the VCMH (4.8 ± 2.1 vs. 7 [IQR 4–10], *p* = 0.250) were similar in subjects with and without thrombosis. Enoxaparin (91%) was the most common anticoagulant prescribed in both groups. The overall median time of thromboprophylaxis was 16.5 days (IQR 12–23), with this being similar between subjects with and without a thrombotic event (18 [IQR 15–26] vs. 17 [IQR 12–23], *p* = 0.465) (Table 2). The characteristics of subjects according to their monthly household income can also be analyzed in Appendix A.

In the bivariate analysis, thromboembolism did not show a statistical association with a low study level or a low monthly household income. However, being Spanish (OR 5.3, 95% CI 1.4–19.5, *p* = 0.005), having a history of VTE (OR 8.1, 95% CI 2.2–28.6, *p* = 0.005), and being a current smoker (OR, 4.7, 95% CI 1.3–16.0, *p* = 0.010) were factors associated with thrombosis (Table 2). In addition, compared with those living in overcrowding, a higher odds ratio of thrombosis was found in subjects that cohabitated with individuals with a high risk for severe COVID-19 (OR 4.7, 95% CI 1.4–16.2, *p* = 0.014). On the contrary, in the adjusted multivariate analysis, none of the variables were associated with thrombosis (Table 2).

## 4. Discussion

The present study has investigated the thrombotic events presented in subjects with COVID-19 who completed isolation in a medicalized hotel during the first wave of the COVID-19 pandemic in Spain. This cohort comprised a group of socially disadvantaged Spanish and migrant populations with poor housing conditions whose monthly household income was, in most cases, lower than EUR 1900. Previous studies have found an association between racial disparities and thromboembolism [14,15]. For instance, significant inequalities in clinical outcomes have been reported in people with PE and socio-economic disadvantages [12]. However, to the best of our knowledge, this is the first report exploring the occurrence of thrombosis in patients with COVID-19 admitted to a temporary medical facility launched for isolation and quarantine. Our results demonstrated that, as observed in other populations, most thrombotic events occurred within the first 90 days after COVID-19 diagnosis and during hospitalization. Moreover, 50% of the population with thrombosis had a monthly household income lower than EUR 950; however, thrombosis was not associated with the socio-economic backgrounds of subjects (housing conditions, monthly household income, and study level).

The relationship between a history of VTE and thrombosis in COVID-19 patients has been previously described [16]; thus, one of the factors that may have played a significant role in the association between Spanish and thrombosis was that seven of the eight subjects with a previous VTE were from Spain. Interestingly, the odds of thrombosis were 4.7 times higher in current smokers compared with former and non-smokers, coinciding with previous reports that indicate that, despite controversial findings, current, former, and heavy smokers have an elevated risk for thrombotic events [17,18,19,20,21]. The risk of thrombosis was also higher among patients cohabitating with individuals with an increased risk for severe COVID-19 compared with those living in overcrowding, which could be associated with family lifestyle-related risk factors for COVID1-9. It is noteworthy that after running a multivariate analysis, none of these factors were associated with thromboembolism.

Material deprivation might increase the likelihood of worse COVID-19 outcomes; however, our study has shown that the cases of thrombosis were not more likely in socio-economically disadvantaged individuals. It must, nonetheless, be noted that our study has some limitations. Firstly, the monthly household income and the level of education of subjects were gathered from a small subgroup of patients, making it difficult to determine the external validity of our findings. In addition, the small number of observed thrombotic events could have impeded obtaining significant results. Finally, the analyses that we performed only included intra-cohort comparisons. Thus, the real impact of our patients’ vulnerability could have been more accurate if a cohort with similar characteristics who did not quarantine in a medicalized facility had been included.

## 5. Conclusions

The study of the contributing factors of COVID-19 thrombosis should include, besides the clinical and personal characteristics of subjects, the analysis of SDOH, such as living conditions and material deprivation. This study has investigated the incidence of thrombosis in a socioeconomically disadvantaged population severely affected during the Spanish lockdown in 2020. Although no association was found between thrombosis and socio-economic factors, our sample might not be large enough to conclude that material deprivation is not associated with thromboembolic events; therefore, further investigation must be performed to study the general socio-economic and environmental conditions of subjects with COVID-19 who develop complications such as thrombosis.

## Figures and Tables

**Figure 1 ijerph-19-07816-f001:**
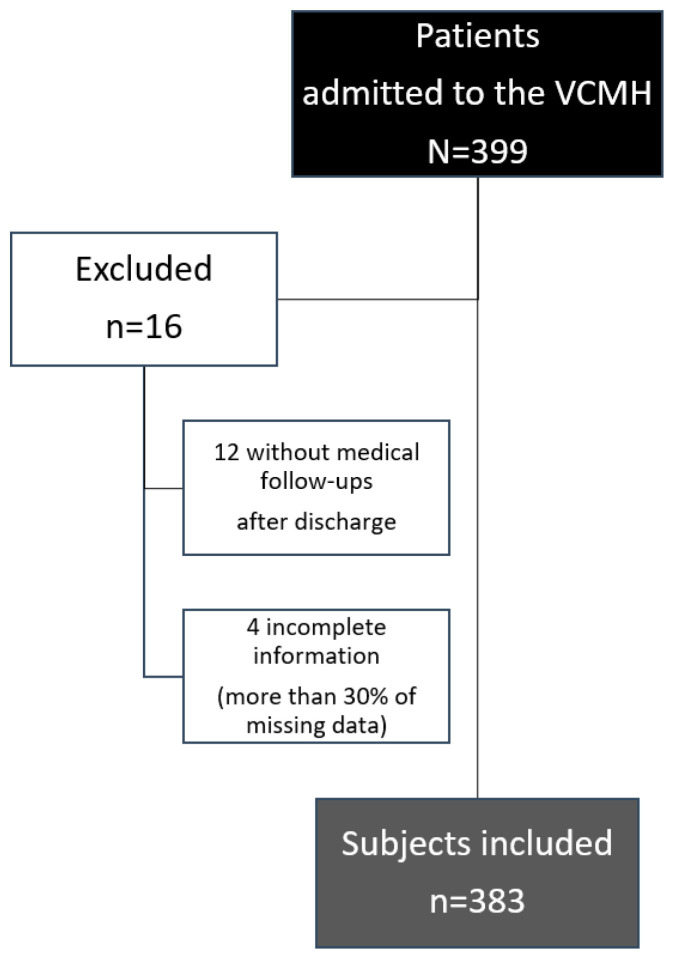
Flowchart of the patients included in the study.

**Figure 2 ijerph-19-07816-f002:**
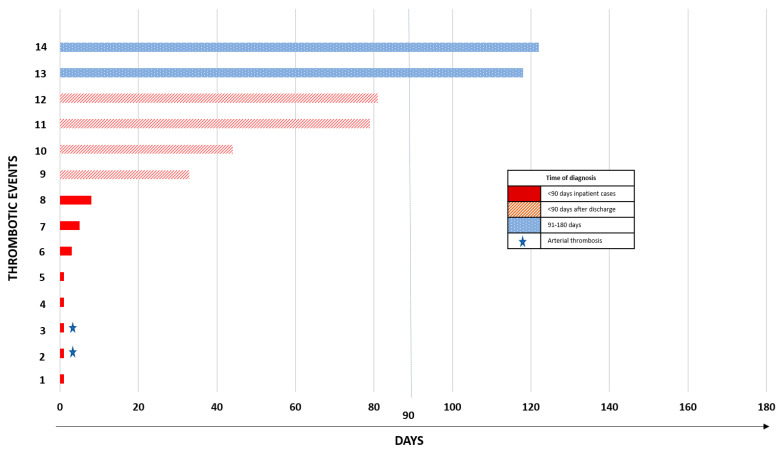
Thrombotic events over the course of 180 days after COVID-19 diagnosis.

**Table 1 ijerph-19-07816-t001:** Characteristics of migrant and Spanish individuals.

	Spanish *n* = 161	Migrants *n* = 222
	% (*n*)	% (*n*)
Sex		
Male	74 (119)	65 (145)
Female	26 (42)	35 (77)
Age, years	62.6 ± 15.3	50.2 ± 11.6
Socio-economic characteristics		
Study level		
Elementary studies	13 (21)	7 (15)
Highschool studies	17 (27)	27 (60)
University studies	9 (14)	6 (13)
Missing	61 (99)	60 (134)
Monthly household income		
< 950 €	9 (14)	18 (40)
Between 950 and 1900 €	21 (33)	14 (32)
Between 1900 and 2850 €	1 (2)	1 (2)
> 2850 €	4 (7)	--
Missing	65 (105)	67 (148)
Reason for referral to the VCMH		
Living in overcrowding	44 (70)	69 (154)
Cohabitating with individuals with an increased risk for severe illness from COVID-19	25 (40)	10 (21)
Homeless, evicted, or shelter residents	4 (7)	8 (18)
Undetermined	27 (43)	13 (29)
COVID-19 management		
Length of hospital stay	9 (IQR 5–12)	8 (IQR 5–13)
Length of VCMH stay	7 (IQR 4–11)	9 (IQR 6–13)
Prophylactic anticoagulation		
Enoxaparin	86 (139)	95 (211)
Bemiparin	11 (18)	2 (5)
Fondaparinux	1 (1)	---
Unknown	2 (3)	3 (6)
Duration of thromboprophylaxis, days	17 (IQR 13–23)	16 (IQR 11–22)
Current smoker	25 (40)	13 (29)
History of alcohol abuse	9 (14)	6 (14)
Pre-existing comorbidities		
Asthma	11 (17)	10 (22)
Chronic pulmonary obstructive disease	6 (9)	1 (2)
Obesity	27 (43)	19 (42)
Dyslipidaemia	44 (71)	35 (77)
Hypertension	44 (71)	19 (41)
Heart disease	22 (35)	4 (8)
Atrial fibrillation	9 (15)	2 (4)
Ischemic heart disease	7 (11)	--
Other heart diseases	6 (9)	2 (4)
Diabetes	17 (27)	7 (15)
History of thrombotic events	7 (11)	3 (7)
VTE	4 (7)	0.4 (1)
PE	1.2 (2)	0.4 (1)
Stroke	1.2 (2)	2 (5)
History of cancer	14 (22)	6 (12)
Pregnancy	1.2 (2)	2 (5)
Mental health disease	30 (48)	24 (53)
Daily medications		
Anticoagulant drugs	13 (21)	1.5 (3)
Antiaggregant	9 (15)	4 (9)
Oral contraceptives	0.6 (1)	3 (7)
Antineoplastics	4 (6)	1.4 (3)

**Table 2 ijerph-19-07816-t002:** Comparison between patients with and without thrombosis.

	Thrombosis *n* = 14 % (*n*)	No Thrombosis *n* = 369 % (*n*)	Crude OR (95%CI)	*p*-Value	Adjusted OR * (95%CI)	*p*-Value
Gender						
Male	71 (10)	69 (254)				
Female (1)	29 (4)	31 (115)	1.1 (0.38–3.6)	0.54	2.1 (0.2–181)	0.735
Origin						
Spanish	79 (11)	41 (150)				
Migrants (1)	21 (3)	59 (219)	5.3 (1.4–19.5)	0.005	3.7 (0.06–224)	0.52
Reasons for referral to the VCMH						
Cohabitating with individuals with a high risk for severe COVID-19	43 (6)	59 (219)				
Living in overcrowding (1)	36 (5)	15 (55)	4.7 (1.4–16.2)	0.014	--	--
Being homeless, evicted, shelter residents	0 (0)	7 (25)	1.02 (1.0–1.04)	0.58	--	--
Studies level						
Elementary studies (1)	7 (1)	9 (35)				
Highschool studies	21 (3)	23 (84)	0.9 (0.09–9.6)	0.72	--	--
University studies	14 (2)	7 (25)	0.3 (0.031–4.1)	0.39	--	--
Monthly household income						
<EUR 950 (1)	21 (3)	14 (51)				
Between EUR 950 and 1900	14 (2)	17 (63)	1.8 (0.2–11.5)	0.41	--	--
Between EUR 1900 and 285	0 (0)	1 (4)	1.0 (0.9–1.1)	0.80	--	--
>EUR 2850	7 (1)	2 (6)	0.10 (0.011–1.06)	0.39	--	--
Comorbidities (reference category: no)						
Current smoker	29 (4)	7 (27)	4.7 (1.3–16.0)	0.010	8.4 (0.00–300)	0.99
History of alcohol abuse	21 (3)	7 (25)	3.7 (0.97–14.6)	0.75	0.7 (0.006–80.6)	0.88
Obesity	36 (5)	22 (80)	2.1 (0.6–6.6)	0.16	0.11 (0.002–7.9)	0.31
Hypertension	43 (6)	29 (106)	1.8 (0.6–5.4)	0.20	1.5 (0–04–49.2)	0.88
Dyslipidaemia	36 (5)	39 (143)	0.86 (0.2–2.6)	0.51	0.23 (0.003–19.4)	0.51
Diabetes	0 (0)	11 (42)	0.95 (0.93–0.98)	0.18	14.2 (0.0001–>1000)	0.99
Heart disease	21 (3)	11 (40)	2.2 (0.5–8.2)	0.20	49.7 (0.001–>1000)	0.97
Asthma	14 (2)	10 (37)	0.69 (0.15–2.9)	0.43	--	--
COPD	(0)	3 (11)	0.96 (0.94–0.98)	0.67	--	--
History of cancer	21 (3)	8 (31)	2.9 (0.78–11.1)	0.12	0.16 (0.001–19.9)	0.45
History of VTE	29 (4)	4 (14)	8.1 (2.2–28.6)	0.005	1.2 (0.0001–>1000)	0.99
Pregnancy	7 (1)	2 (6)	4.5 (0.5–40.9)	0.23	--	--
Mental health disease	14 (2)	27 (99)	0.44 (0.09–2)	0.23	8.3 (0.0001–>1000)	0.99

* The multivariate analysis was adjusted by the variables: origin, cohabitating with individuals with an increased risk of severe COVID-19, being a current smoker, and having a history of VTE. (1) = reference category.

## Data Availability

The authors cannot share the data used for this analysis because it is not publicly available. However, it can be shared upon formal request to the patient blood management research group of the Research Institute of the University Hospital La Paz, C. de Pedro Rico, 6, 28029 Madrid.

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
