# Peer review of "Thromboembolic Events in a Socio-Economically Disadvantaged Population with COVID-19 Admitted to a Medicalized Hotel in Madrid"

_ijerph, 2022, doi:10.3390/ijerph19137816_

Round 1
Reviewer 1 Report
The article was well written. But the objective of this study was overly humble. I have a question for the author's. How many of the volunteers who took part in this study have been vaccinated? If the author's can show more details related to the ongoing vaccination program, the results of this investigation would be more useful to the readers. I hope the authors can further comment on this issue.
Author Response
Dear revisor,
We appreciate the time you have taken to revise our manuscript and your valuable suggestions. Regarding the question concerning whether the population of our study underwent a vaccination program, we have clarified that due to patients were attended at the medicalized hotel during the first wave of COVID-19 in Spain (March-May 2020), and they were followed up for a 180-days period after COVID-19 diagnosis (up to November 2020), any patient had been vaccinated at the time of this study. This information has been added to the manuscript in lines 80-83.
Reviewer 2 Report
The Authors decided to analyze the impact of social aspects of the COVID-19 complications. The idea and main results are very interesting, however I would like to express my concerns and comments about the study.
Please provide AMI criteria / AMI type that was used for including AMI to arterial thrombosis events.
There are two arterial thrombosis events mentioned in the paper - what were they exactly?
Please provide overcrowding definition.
Which was the low income cut-off? I assume that below 950 euros but please make it clear in the manuscript.
Please add characteristics of the low/high income populations as it is the subject of the analysis (analogically to Table 1)
Please add p value to all the results.
Line 166: did you mean 900 or 950 euros?
Author Response
Point 1: Please provide AMI criteria / AMI type that was used for including AMI to arterial thrombosis events.
Response. The criteria for arterial thrombosis were set based on the food and drug administration sentinel protocol “study synopsis: natural history of coagulopathy in COVID-19”1. Further details regarding the ICD-10-CM and ICD-9-CM classifications that were considered for acute myocardial infarction can be found in the following link:
- https://www.sentinelinitiative.org/sites/default/files/documents/Coagulopathy_COVID19_Study_Synopsis_v2.0_0.pdf
Attending this comment, we have included the reference [5] in line 87.
Point 2: There are two arterial thrombosis events mentioned in the paper - what were they exactly?
Response. We appreciate your question. One case of iliac artery thrombosis and a case of middle cerebral artery stroke were found. Considering this point, we have included this information in lines 139-140.
Point 3. Please provide overcrowding definition.
Response. Overcrowding was considered if a household crowding index (HCI) ≥2.5. The HCI is calculated as the total number of co-residents per household, excluding newborn infants, divided by the total number of rooms, excluding the kitchen and bathrooms.
Considering this observation, we have included the definition of overcrowding in lines 99-102.
Point 4. Which was the low-income cut-off? I assume that below 950 euros but please make it clear in the manuscript.
Response. Yes, the minimum interprofessional wage in Spain in 2020 (the time in which the household income was measured in our study), was 950€. We have clarified it in our manuscript in line 127.
Point 5: Please add characteristics of the low/high-income populations as it is the subject of the analysis (analogically to Table 1).
We have included the characteristics of the subjects according to their monthly household income in appendix 1.
Point 6: Please add a p-value to all the results.
We have included the p-values of the results in Table 2.
Point 7: Line 166: did you mean 900 or 950 euros?
Response. Yes, we meant 950 euros, we have changed it.
Reviewer 3 Report
This is an interesting study, worth of publishing
Author Response
Dear revisor,
We appreciate your comment and your consideration for accepting our manuscript for publication.
Kind regards.
Reviewer 4 Report
Dear authors,
Thrombosis is one of the most significant complication in COVID-19, as well as many other conditions. Identifing the risk factors for this is so of outmost importance. For this reason, the research is an important contribution to the field. However, I have some remarks regarding the manuscript:
1. The text os full of grammar mistakes, some of them I mentioned in the attached pdf, please ask for the help of a profesional English editor to correct the whole manuscript.
2. In aTable 1 there are items in which the total number of the subjects is not 383, please check once again, correct or otherwise explain.
3. Table 2 needs proper formating and the p value needs to be mentioned.
4. I recomend also a multivariate analysis in order to identify the independent risk factors for thrmbosis in this particular population.
5. Please explain why authors think that diabetes and COPD were protective factors for developing TE.
6. The major limitation of the study I think is the low number of TE events in this population, thus impeding to obtain significant results for some risk factors.

Author Response
Dear revisor.
We deeply appreciate your suggestions and corrections and we thank you for the time that you have taken to extensively revise our manuscript.
Point 1. The text of full of grammar mistakes, some of which I mentioned in the attached pdf, please ask for the help of a professional English editor to correct the whole manuscript.
We appreciate the corrections that you have made. We modified the errors that were highlighted in the text and an English editor has reviewed our manuscript.
Point 2. In Table 1 there are items in which the total number of the subjects is not 383, please check once again, correct, or otherwise explain.
We have included the missing values.
Point 3. Table 2 needs proper formatting and the p-value needs to be mentioned.
Table 2, has been formatted and the p values have been introduced
Point 4. I also recommend a multivariate analysis in order to identify the independent risk factors for thrombosis in this particular population.
A multivariate analysis adjusted by the variables that resulted to be significant in the bivariate analysis was added to Table 2.
Point 5. Please explain why the authors think that diabetes and COPD were protective factors for developing TE.
This association was finally not statistically significant in the multivariate analysis.
Pont 6. The major limitation of the study I think is the low number of TE events in this population, thus impeding obtaining significant results for some risk factors.
This limitation has been included in the text.
Round 2
Reviewer 1 Report
In general, the article was well written. However, the conclusion of this study is too controversial. The link between the socioeconomic status and the thromboembolism has been very much statistically established in other countries before and during the COVID pandemic [1-5]. The conclusion of this study has clearly contradicted those of other similar investigations.
There must be a reason for this contradiction. The authors should comment on all the possible causes leading to this contradiction. Is the sample size large enough to draw such a conclusion? Since the study was completed before the commencement date of the Spanish vaccination program, is it arguable that the follow-up time of this study was too short to draw such a conclusion?
1) Jørgensen H, Horváth-Puhó E, Laugesen K, Braekkan S, Hansen JB, Sørensen HT., "Socioeconomic status and risk of incident venous thromboembolism.", J Thromb Haemost. 2021 Dec;19(12):3051-3061. doi: 10.1111/jth.15523.
2) Kort D, van Rein N, van der Meer FJM, Vermaas HW, Wiersma N, Cannegieter SC, Lijfering WM., "Relationship between neighborhood socioeconomic status and venous thromboembolism: results from a population-based study.", J Thromb Haemost. 2017 Dec;15(12):2352-2360. doi: 10.1111/jth.13868.
3) Alexander T. Cohen, Anne-Céline Martin, and Carlos Martinez, "Epidemiology and socioeconomic consequences of venous thromboembolism", in ESC CardioMed (3 edition), DOI:10.1093/med/9780198784906.003.0655_update_001
4) Isma N, Merlo J, Ohlsson H, Svensson PJ, Lindblad B, Gottsäter A. Socioeconomic factors and concomitant diseases are related to the risk for venous thromboembolism during long time follow-up. J Thromb Thrombolysis. 2013 Jul;36(1):58-64. doi: 10.1007/s11239-012-0858-8. PMID: 23247894.
5) Rishi K. Wadhera, Eric A. Secemsky, Yun Wang, Robert W. Yeh and Samuel Z. Goldhaber, "Association of Scocioeconomic Disadvantage with Mortality and Readmissions Among Older Adults Hospitalized for Pulmonary Embolism in the United States.", Journal of the American Heart Association. 2021, https://doi.org/10.1161/JAHA.121.021117
Author Response
Dear reviewer,
We agree that this controversial finding might be related to the small sample size that we analyzed. Therefore, we have included a statement regarding this limitation in lines 213-214. On the contrary, we do not consider that the follow-up period (180 days) could have been related to this finding.
Reviewer 2 Report
Thank you for the revision of your manuscript.
The Authors have included my suggestions and I believe that quality of the article has much improved.
Nonetheless I would like to note that in Table 2 there still lines with p value missing. Please correct it.
Author Response
Dear reviewer,
We appreciate all the suggestions that you have made, they were so helpful to improve the quality of our manuscript.
We have revised Table 2, and we have introduced the missing p-values. In addition, some percentages have been corrected.
Thank you very much for the time that you have taken to revise our manuscript.
Kind regards